# Predicting the Invasion Potential of the Lily Leaf Beetle, *Lilioceris lilii* Scopoli (Coleoptera: Chrysomelidae), in North America

**DOI:** 10.3390/insects11090560

**Published:** 2020-08-23

**Authors:** Maggie Freeman, Chris Looney, Marina J. Orlova-Bienkowskaja, David W. Crowder

**Affiliations:** 1Washington State Department of Agriculture, 1111 Washington St. SE, Olympia, WA 98504, USA; clooney@agr.wa.gov; 2Department of Entomology, Oregon State University, Hood River, OR 97031, USA; 3A.N. Severtsov Institute of Ecology and Evolution, Russian Academy of Sciences, 33 Leninskii pr., Moscow 119071, Russia; marinaorlben@yandex.ru; 4Department of Entomology, Washington State University, Pullman, WA 99164, USA; dcrowder@wsu.edu

**Keywords:** lily leaf beetle, maxent, habitat suitability model, lilies, invasive insect

## Abstract

**Simple Summary:**

The lily leaf beetle, *Lilioceris lilii* Scopoli (Coleoptera: Chrysomelidae), is an invasive pest of cultivated and native lilies (Liliaceae). This Eurasian beetle was introduced to North America in 1943, and can now be found in nine provinces and 14 states. The lily leaf beetle has already been found feeding and reproducing on several eastern species of native lilies. To help predict where *L. lillii* will be able to establish in North America and which native Liliaceae species will be vulnerable to attack, a habitat suitability model was created. This model uses specimen location data along with environmental data to produce habitat suitability estimates between 0 (low suitability) and 1 (high suitability). Model results indicate that the beetle should be able to establish throughout the range of most North American Liliaceae, including species of special conservation concern. With the increased utilization of habitat suitability models in conservation management, this model should be consulted, to help plan preemptive monitoring and control efforts for *L. lilii* in regions, with Liliaceae species of conservation concern.

**Abstract:**

Invasive species are among the leading threats to global ecosystems due to impacts on native flora and fauna through competition and predation. The lily leaf beetle, *Lilioceris lilii* Scopoli (Coleoptera: Chrysomelidae), is an invasive pest of lilies (*Lilium* spp.) and other genera of Liliaceae (Liliales). A habitat suitability model was created using Maxent, to help predict if *L. lilii* will be able to establish in locations were native North American Liliaceae species grow. The model was created using georeferenced occurrence records from the beetle’s native, naturalized, and invasive range. Model results indicate that precipitation in the driest quarter and annual average temperatures are most strongly correlated with *L. lilii* distribution, and suggest that the species will perform poorly in very dry, hot, or cold environments. The model also indicates that the beetle should be able to establish throughout the range of most North American Liliaceae genera, including species of special conservation concern. This model can be used by natural area managers to identify areas of high habitat suitability that overlap with vulnerable North American Liliaceae species, and prioritize *L. lilii* monitoring and control activities as the beetle continues to expand its range.

## 1. Introduction

Invasive species are among the leading threats to global ecosystems, due to impacts on native flora and fauna through competition and predation [1,2,3]. The number of invasive plant and animal species introduced to new areas has increased with the expansion of global trade [4,5,6], with horticultural trade alone responsible for the establishment of 42 new insect species in the USA between 1997 and 2001 [4]. This number will likely increase as climate change modifies environmental conditions, allowing alien species to establish in areas where they were previously unable to survive [1,4]. In the last two decades, the development and use of habitat suitability models have proliferated, creating tools that can help highlight the geographical regions likely to be suitable to invasive plant and animal species [7,8,9,10]. These models are now being utilized by land managers to predict the range and impacts of introduced plant and animal species, anticipate potential threats, and enact preemptive monitoring or control activities to mitigate the impacts of new invaders [10,11,12,13].

The lily leaf beetle, *Lilioceris lilii* Scopoli (Coleoptera: Chrysomelidae), is a Eurasian beetle that feeds on species of Liliaceae (Liliales), primarily *Lilium* spp., and *Fritillaria* spp. Larvae and adults are voracious feeders, and can destroy native and cultivated lilies in places where they lack control. Defoliation is frequently complete, causing immediate loss of aesthetic value, and even plant death within a few years [14]. In Europe *L. lilii* has been documented feeding on 57 species of hybrid *Lilium*, 30 species of *Lilium*, 5 species of *Fritillaria* and one species of *Cardiocrinum* (Liliales: Liliaceae) [15]. The oldest record of the beetle in North America was from the Northwest Territory [16], but there is no evidence that the species became established after this apparent early introduction [17]. The beetle was found again in North America, in Montreal, Canada, in 1943 [18], where it was successfully established. After several decades, the beetle began to spread rapidly, appearing in multiple northeastern states and provinces throughout the early 1990s [19]. The beetle continues to expand its range, with records now from nine provinces and 14 states [20]. The beetle became established in the Pacific Northwest by 2011, and has since continued to spread throughout the Puget Sound [21,22].

Although *L. lilii* prefers cultivated Asiatic and Oriental lilies, it has been documented feeding on several eastern species of native North American *Lilium*, *Lilium canadense* L., and *Lilium michiganense* Farwell [23,24]. The beetle is also able to complete its lifecycle on rosy twisted stalk, *Streptopus lanceolatus* (Aiton) (Liliales: Liliaceae), a newly documented host genus for the beetle [19].

Freeman et al. [25] conducted feeding trials on several western species of Liliaceae, demonstrating that *L. lilii* larvae readily fed upon and completed their lifecycles on *Lilium columbianum* Leichtelin. The beetle was also able to complete its lifecycle on *Calochortus tolmiei* Hooker and Arnott (Liliales: Liliaceae), although survival rates were low. *Calochortus* is restricted to western North America, and includes 10 species of conservation concern across the western states [26]. There are an additional 25 species of conservation concern in the other genera of western Liliaceae, found in habitats ranging from high desert to moist forests [26]. *Lilioceris lilii* could pose an additional threat to these many at-risk species, raising questions about the beetle’s ultimate range as it continues to spread in North America.

The broad distribution of *L. lilii* in Asia and Europe suggests that it can establish itself throughout much of North America [14,15,19,23,27,28]. To test this prediction, we constructed a species distribution model using maximum entropy with Maxent v3.4.1 [29]. Maxent utilizes presence-only specimen data in conjunction with environmental data (e.g., annual mean temperature), to produce habitat suitability estimates between 0 (low suitability) and 1 (high suitability). The model works by finding the distribution within the defined study area that is closest to geographically uniform (maximum entropy), based on environmental conditions found at recorded species occurrence locations [30]. We selected the Maxent modeling approach for this analysis because of its use of presence-only data, and its consistently high predictive performance compared to other modeling methods [31,32]. Models that use presence-absence data require records acquired through systematic biological surveys, which generally do not exist for *L. lilii*. We then used outputs from the model to compare potential *L. lilii* range with known locations of western Liliaceae, to identify the plant species most at risk from this invasive pest.

## 2. Materials and Methods

### 2.1. Occurrence Records

A total of 1392 georeferenced occurrence records were assembled from collaborative *L. lilii* researchers in Canada [33], gleaned from primary literature [34,35,36], or obtained from our own research [22,37,38]. These records represented 632 sites in Europe, 150 sites in Asia, and 610 sites in North America (Figure 1). According to the Catalogue of Paleartic Coleoptera [39], *L. lilii* is recorded from 40 European countries, European Russia, nine Asian countries, and Asian Russia (no occurrence records provided in the publication). We were unable to find georeferenced occurrence records for 11 of the 50 countries in the beetle’s recorded range (5 in Europe, and 6 in Asia).

Occurrence-only data sets often exhibit sampling bias, i.e., higher observation records in areas that are easier to access and survey [40,41]. To reduce sampling bias, the occurrence records were reduced to one per 100 km^2^ [42,43]. This was accomplished by joining all sites with a 100 km^2^ vector grid in QGIS and randomly selecting one site per 100 km^2^ grid, leaving 466 occurrence records (297 in Europe, 75 in Asia, and 94 in North America). This also ensured that random background points used to train the model were more evenly represented across Europe, Asia, and North America [43].

### 2.2. Bioclimatic Variables

Nineteen bioclimatic variables were downloaded from the Worldclim Database [44], at a spatial resolution of 30 arc seconds (~1 km^2^). These bioclimatic variables were then tested for high multicollinearity, to avoid over-fitting subsequent models [45,46]. After running a variance inflation factor analysis, following the methods outlined in Pradhan (2016) [47], we selected 11 biologically meaningful bioclimatic variables. These were identified by running a Pearson correlation coefficient analysis, using the 19 bioclimatic values at all occurrences of *L. lilii* in Asia, Europe, and North America. After obtaining Pearson correlation coefficient “r” values, we derived “R2” values (r × r), and VIF values, using the formula [48]:[1/(1 − r2)]

Variables that had Pearson correlation “r” and “R2” values of >0.8, and VIF values of >10 were examined. The less informative variable in the correlated pairs examined was removed, leaving 11 bioclimatic variables. These variables were: Annual Mean Temperature (BIO1), Mean Diurnal Range (Mean of monthly (Max temp-Min temp)) (BIO2), Isothermality (Mean Diurnal Range/Temperature annual range) (*100) (BIO3), Mean Temperature of Wettest Quarter (BIO8), Mean Temperature of Driest Quarter (BIO9), Mean Temperature of Warmest Quarter (BIO10), Precipitation Seasonality (BIO15), Precipitation of Wettest Quarter (BIO16), Precipitation of Driest Quarter (BIO17), Precipitation of Warmest Quarter (BIO18), and Precipitation of Coldest Quarter (BIO19).

### 2.3. Species Distribution Modelling in Maxent

Rather than using the classical approach of projecting from the native range onto the invaded range, we chose to run the model using all locations where this beetle has been found in North America, Europe and Asia. Some studies have shown that using projection to predict invasion onto new environments can underpredict suitable habitat in the invaded range [49,50]. For example, species that are limited to certain areas by niche competition or by natural enemies in their native range may be able to invade new climatically distinct areas in their invaded range [51,52,53,54]. These studies obtained more realistic models by combining location data from both the native and invaded ranges [49,50]. This approach was supported by early iterations in which we trained the model on occurrence records from Eurasia, and used the projection function to predict habitat suitability for North America. These results showed poor habitat suitability in many of the states and provinces where *L. lilii* was already well established.

Because Maxent works under the assumption that the species is at equilibrium with its environment, and that the species environmental niche is conserved across space and time [55,56], we included a sampling bias file that instructs Maxent to only draw background samples from Asia, Europe, and the invaded regions (counties and municipalities) in North America, where the presence of the beetle has been confirmed. Since the lily leaf beetle was only recently established in North America (1943) and its range continues to spread, this species has not had time to invade all potentially suitable locations on the continent. By using the bias layer, we prevented the model from drawing pseudo-absence points from potentially suitable, but as yet unencountered, areas of North America [40,41].

The model was executed using default settings for prevalence, feature types, complementary log-log output and regularization [30]. Under the settings, the number of iterations was increased from 500 (default) to 5000, to provide extra time for convergence.

To evaluate model performance, we produced 50 replicate model runs, in which we used the Maxent cross-validation method “Subsample” training on 75% of the data, and testing on the remaining 25% [41]. Model fit was evaluated by the area under the curve statistic (AUC), where values approaching 1 are indicative of a strong model fit, values around 0.50 approximate a model that is equivalent to random, and values approaching 0 indicate a model that performs worse than random [41,57].

### 2.4. Liliaceae Site Records

To visualize which native Liliaceae species may be impacted by the spread of *L. lilii*, the habitat suitability model was overlaid with location data for the species of *Calochortus*, *Fritillaria*, *Lilium*, *Medeola* (Liliales: Liliaceae), and *Streptopus* in North America. Specimen records were obtained from the Consortium of Pacific Northwest Herbaria, Consortium of Midwest Herbaria, Consortium of Northeastern Herbaria, and the South-East Regional Network of Expertise and Collections [58]. To evaluate which of these genera are in areas with high invasion potential, we used the point sampling tool in QGIS to determine the habitat suitability score for each Liliaceae site record. We then categorized Liliaceae records, with habitat suitability scores 0.50 or greater having high invasion potential, and Liliaceae records with habitat suitability scores less than 0.50 having low invasion potential. The borders for all states that have endangered, threatened, rare, or sensitive plant species in the genera *Calochortus*, *Fritillaria*, *Lilium*, and *Streptopus* listed on the USDA Natural Resources Conservation Service website were highlighted [26].

## 3. Results

### 3.1. Exploitation of Available and Occupied Bioclimatic Ranges

The variable ‘Precipitation of the Driest Quarter’ (Bio17) had the highest predictive power in the final model (Figure 2). This suggests that water availability during drier periods is crucial for *L. lilii* or is highly correlated with other habitat features that determine the beetle’s ability to survive and establish in an area. ‘Annual mean temperature’ (Bio 1) also plays an important role in determining which habitats the beetle can occupy. Areas predicted to have the highest habitat suitability had between 80 mm and 300 mm cumulative rain fall in the driest quarter, and average annual temperatures between 3 °C and 15 °C.

### 3.2. Predicted Global Distribution Based on All Known Occurrences

The model’s average AUC ± SD was 0.898 ± 0.01 (Figure 3a). Most of the beetle’s native range in Asia and its naturalized range in Europe were correctly predicted by the model. However, Mongolia (Asia), presumably within the beetle’s native range, is shown to be largely unsuitable for *L. lilii*.

For North America, the model predicts that several states and provinces east of the Missouri river have very high habitat suitability for the lily leaf beetle, with the exception of the eastern portion of the Carolinas, Southern Georgia, Florida, and Northern Ontario and Quebec (Figure 4). The southern portions of several north-central provinces and states are also predicted to have high habitat suitability. Habitat suitability drops drastically in the deserts of the Southwest, with only a few areas of low to moderate suitability around the Rocky and Cascade mountain ranges. A moderate to high pocket of habitat suitability can be found in northwestern California, western Oregon and Washington, and along the Rocky Mountain range in British Columbia. The model indicates high to moderate habitat suitability in all of the regions where the lily leaf beetle has been reported in North America (Figure 3b).

When evaluating habitat suitability scores for native Liliaceae specimen records, almost all *Lilium*, *Medeola*, and *Streptopus* species are within the range where *L. lilii* should be able to readily be established (≥0.50) (Figure 4c–e). Only *Lilium* species found in Colorado (USA), Florida (USA), Idaho (USA), Kansas (USA), Nevada (USA), and New Mexico (USA), *Medeola* species found in South Carolina (USA), and *Streptopus* species in Alaska (USA), Arizona (USA), California (USA), Idaho (USA), New Mexico (USA), Utah (USA), and the Yukon (Canada) had habitat suitability scores lower than 0.50. *Calochortus* species found in British Columbia (Canada), California (USA), Colorado (USA), Montana (USA), Oregon (USA), South Dakota (USA), Washington (USA), and Wyoming (USA) were in areas with habitat suitability scores ≥ 0.50 (Figure 4a). *Fritillaria* species found in Alaska (USA), British Columbia (Canada), California (USA), Massachusetts (USA), Montana (USA), Oregon (USA), South Dakota (USA), Washington (USA), and Wyoming (USA) were in areas with habitat suitability scores ≥ 0.50 (Figure 4b).

Habitat suitability predictions indicate that the lily leaf beetle will be able to establish in all states (USA), with endangered, threatened, rare, or sensitive *Calochortus*, *Fritillaria*, *Lilium*, *Medeola,* and *Streptopus* species, with the exception of Florida (Figure 4f).

## 4. Discussion

This study illustrates that *L. lilii* has a potentially broad distribution in North America, and will likely continue to expand its range. This analysis also identifies areas with native plants of conservation concern that may be vulnerable to attack by *L. lilii*. The output from this model can inform efforts by resource managers to predict which native plant species could be affected as the beetle expands its range.

The lily leaf beetle’s distribution across Eurasia indicates that this insect can survive in a wide range of climatic conditions in North America, with the exception of cold and dry environments. This is reflected in the model output, which indicates that arid habitats and areas with average annual temperatures below −10 °C and above 20 °C are generally unsuitable for the beetle. The model suggests that *L. lilii* will be able to generally establish between 40–50-degrees latitude across North America.

These results can be used to identify sensitive habitats for monitoring, as *L. lilii* naturally migrates or is artificially spread into new areas. All states except Florida with at-risk species of *Calochortus*, *Fritillaria*, *Lilium*, *Medeola*, and *Streptopus* have regions with moderate to high habitat suitability for the lily leaf beetle. Expansion of the beetle into natural areas may pose significant challenges to land managers since control options for the beetle are limited. Because *L. lilii* feeds on flowering plants, the use of broad-spectrum pesticides in wild habitats is problematic, due to toxicity to pollinators or other non-target species. Hand removal and destruction of larvae, eggs, and adults can be successful on small scales, but there is no appreciable likelihood of hand control in wild habitats. Biopesticides (e.g., neem (azadirachtin)) used as an alternative to chemical control mechanisms must be reapplied frequently as new larvae hatch. The most promising control option in North America is via three parasitoid wasp species, *Tetrastichus setifer* Thomson (Hymenoptera: Eulophidae), *Lemophagus errabundus* Szepligeti (Hymenoptera: Ichneumonidae), and *Diaparsis jucunda* (Holmgren) (Hymenoptera: Ichneumonidae), released as part of a classical biological control program [59]. Releases have been made in several eastern states and provinces, and in Washington State [25,60]. While these agents provide an environmentally safe, long-term control option for the beetle in natural ecosystems, they disperse very slowly, and can be more quickly deployed through deliberate release. The results of this model may help managers prioritize natural areas for beetle monitoring and speedy parasitoid release upon discovery.

While the results of this model are robust, its veracity could be limited by sampling bias in the underlying data, the beetle’s continuing range expansion, and impacts of future climate change [40,41,49,50,61]. For example, *L. lilii’s* natural range comprises the temperate latitudes of East Asia, encompassing Kazakhstan, Mongolia, northern China, and South of East Siberia [37], but occurrence records for the region are rare. This could indeed reflect limited suitability, expressed as low beetle abundance or geographically restricted habitat. This lack of native records could also be due to sampling bias and the less apparent nature of the beetle in its native range. In its native range, it feeds on wild lilies, faces competition from other species, and has multiple natural enemies [27,28,42,62]. These factors likely limit population sizes and visible damage, contributing to infrequent observations. In Europe and North America, where the beetle predominantly feeds on cultivated lilies, gardeners are frequently confronted with this pest, and are more likely to report it [42,63]. The lack of specific occurrence records for the beetle in Mongolia and limited records in northern China may compromise the model’s output and the accuracy of habitat suitability estimates for these regions in Asia and similar habitats in North America. This is especially true if the missing records occur in regions with extreme values for annual temperature and precipitation in the driest quarter. Specific locality records from these regions might allow more accurate modelling and reveal a more expansive predicted habitat.

Maxent operates under the assumptions that the species is at equilibrium and the environmental niche of the species is conserved across space and time [56]. These assumptions can be violated when modelling an invasive species with an expanding range that may be able to thrive in climactic and ecological conditions unrepresented in their native range [51,52,53,54]. There are relatively few lily feeding insects in North America, there are no native predators that regularly attack the beetle, and there are new host plants that the beetle can utilize in natural areas [19,27,64]. *Lilioceris lilii’s* realized niche in North America may be more expansive than in Eurasia, and this model may not be able to highlight some potentially suitable habitat on the continent.

Interestingly, the beetle’s range in European Russia is also spreading into the north and northeast. In the last two decades, the pest appeared in Vladimir, Ivanovo, Kirov, Kostroma, Nizhnij Novgorod, and Yaroslavl regions, and in Udmurtia and Chuvashia [43]. This range expansion could be connected with climate change. The annual mean temperature for Russia has risen by 1.29 °C over the past century, and is projected to rise 1.1 ± 0.5 °C by 2020 and 2.6 ± 0.7 °C by 2060 [65]. The winter mean surface temperature is projected to increase 3.4 ± 0.8 °C by 2060 [65]. Overall mean global temperatures have increased by approximately 1 °C, and further accelerated warming is predicted [66]. Increasing evidence indicates that global warming has caused species to shift their range to more poleward latitudes and higher elevations [67,68,69,70,71,72], helping alien species expand into regions where they previously were unable to survive and reproduce [73,74,75]. Low temperatures are key constraints on the range of many insect species, but many northern expansions have already been recently documented [75]. Our results indicate that warmer winter temperatures and changing precipitation regimes are likely to have a large effect on *L. lilii’s* future distribution, which may be exacerbated by a concomitant expansion of cultivated and wild host material [76]. As new locations become suitable for growing lilies, the anthropogenic movement of this pest will also increase.

With the increased utilization of habitat suitability models in conservation management, this model should be consulted to help plan preemptive monitoring and control efforts for *L. lilii* in regions with lily species of conservation concern. As new location data become available for *L. lilii*, this model should be revisited to provide the most comprehensive prediction of the beetle’s distribution and its potential impact on native Liliaceae species.

## 5. Conclusions

Our model results indicate that the lily leaf beetle will be able to expand its range into many new regions of North America. These results highlight that most endangered, threatened, rare, or sensitive *Calochortus*, *Fritillaria*, *Lilium*, *Medeola* and *Streptopus* species in the United States could be invaded and further impacted by the lily leaf beetle as it expands its range.

## Figures and Tables

**Figure 1 insects-11-00560-f001:**
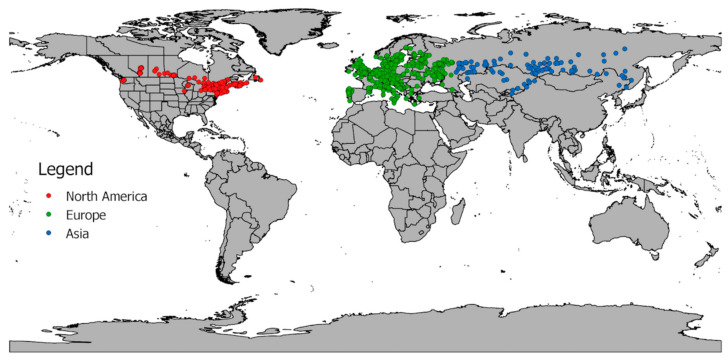
Current global distribution of *Lilioceris lilii*. Current distribution of *L. lilii* obtained from research collaborators and published literature. Blue dots represent occurrences in Asia; green dots represent occurrences in Europe, and red dots represent occurrences in North America.

**Figure 2 insects-11-00560-f002:**
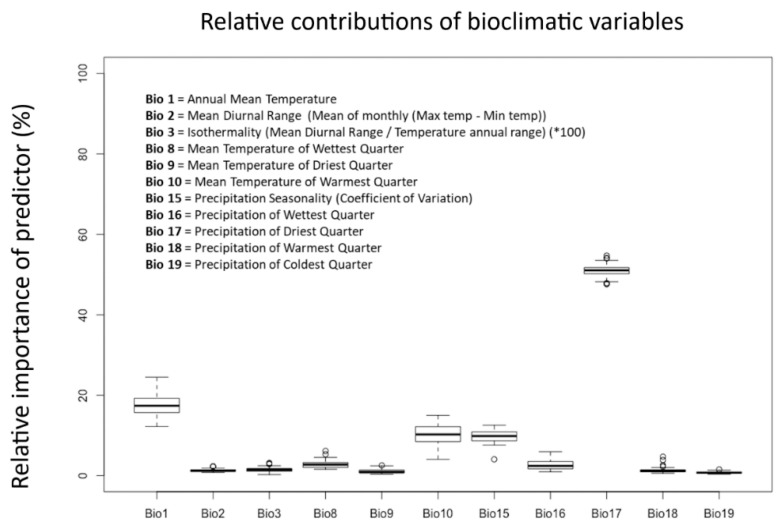
Relative contributions of bioclimatic variables. Boxplots showing the relative contributions of 11 bioclimatic variables to the Maxent model. The plots show variation in the importance of each predictor within the 50 replicate runs for the model. Below is a legend describing what the eleven bioclimatic variables represent in relation to temperature and precipitation.

**Figure 3 insects-11-00560-f003:**
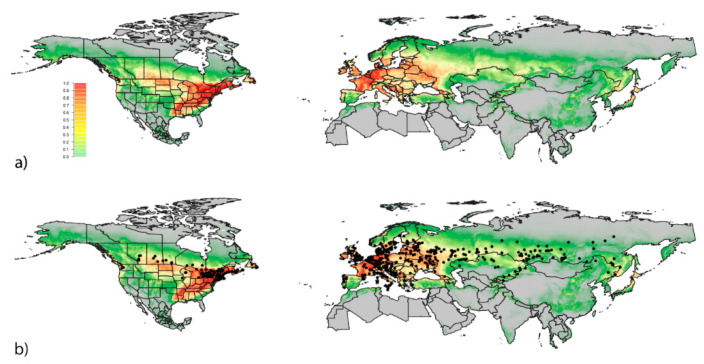
Habitat suitability model of native and invaded range. Predicted probability of occurrence of *Lilioceris lilii*, ranging from lowest (0, grey) to highest (1, red) probability, based on 466 data points given by the “cloglog” output in the Maxent software. The model’s average AUC ± SD was 0.898 ± 0.01. Panels show: (**a**) habitat suitability model of North America, Europe, and Asia (**b**) habitat suitability model overlaid with *L. lilii* site data.

**Figure 4 insects-11-00560-f004:**
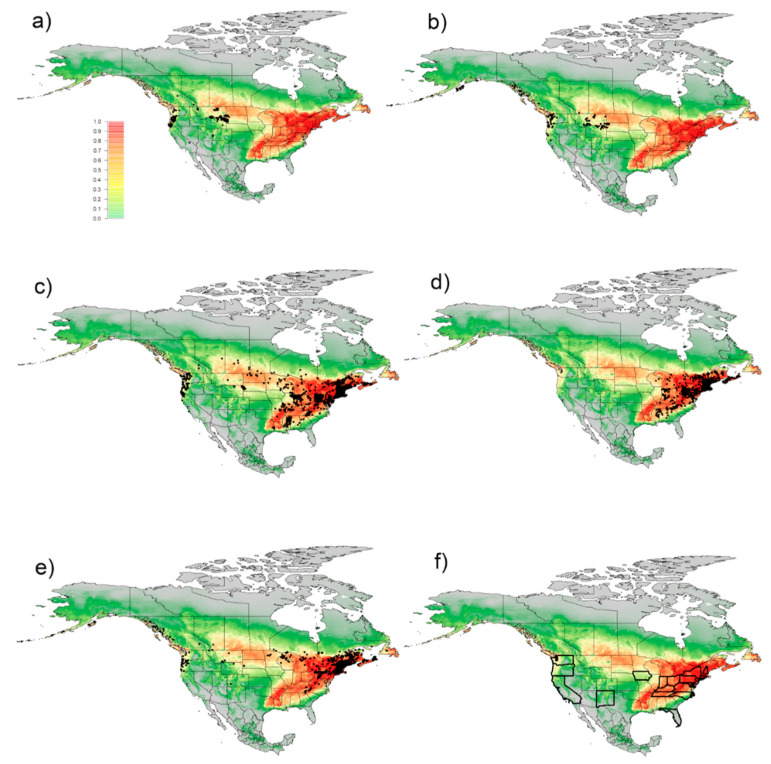
Habitat suitability model overlaid with native Liliaceae records. Habitat suitability model of North America overlaid with site data for native species of *Calochortus*, *Fritillaria*, *Lilium*, and *Streptopus* that had habitat suitability scores of ≥0.50. Predicted probability of occurrence of *Lilioceris lilii* based on 466 occurrences, ranging from lowest (0, grey) to highest (1, red) given by the “cloglog” output in the Maxent software. Panels show: (**a**) *Calochortus* spp. (**b**) *Fritillaria* spp. (**c**) *Lilium* spp. (**d**) *Medeola* spp. (**e**) *Streptopus* spp. (**f**) Borders of states with federally listed endangered, threatened, rare, or sensitive species of *Calochortus*, *Fritillaria*, *Lilium*, and *Streptopus* are in bold.

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
