# Peer review of "Predicting the Invasion Potential of the Lily Leaf Beetle, Lilioceris lilii Scopoli (Coleoptera: Chrysomelidae), in North America"

_insects, 2020, doi:10.3390/insects11090560_

Round 1

Reviewer 1 Report

The paper is based on an analisys of records elaborated with a predictive model to improve the knowledge of the biological potentiality of an invasive species.

I have reported only a few minor comments and corrections in the MS. 

You will have no problem in applying them.

All the best

Author Response

Thank you for providing us feedback on our manuscript. I appreciate how quickly we received your review. After incorporating your suggestions, our article is much clearer and is rid of embarrassing mistakes.  

Below I address each of your suggested revisions. Before each response I added the new line number(s) that the corresponding revision can be found.

Reviewer 1:

  • L3 (title):  "Lilioceris lilii" should be italicized
    • L3: "Lilioceris lilii" was changed "Lilioceris lilii
  • L4: “In North America” the “I” in “In” should be lowercase
    • L4: “In” was changed to “in”
  • L16: “Lilioceris lilii” should be italicized
    • L16: "Lilioceris lilii" was changed "Lilioceris lilii
  • L17: “(Liliaceae.)” the period should be removed
    • L17: “(Liliaceae.)” was changed to “(Liliaceae)”
  • L20: “L. lilii” should be italicized
    • L20: “L. lilii” was changed to “L. lilii
  • L34: Should “precipitation” be changed to “precipitations”?
    • L30: I kept “precipitation” because “precipitation in the driest quarter” is the name of the Worldclim variable name that I am referencing.
  • L47: Author could introduce here the potential influence of climatic changes in offering a larger range of habitat where established, enhancing the considerations reported in discussion.
    • L48-50: A sentence about the implications of climate change allowing invasive species to expand into new regions was added.
  • L88: “absence” should be changed to presence-absence
    • L91: “absence” was changed to “presence-absence”
  • L96: “[22, 37-38](.” Remove parentheses
    • L99: “[22, 37-38](.” Changed to “[22, 37-38].”
  • L249: I think it would be much better to nominate the three parasitoid species directly in the text
    • L271-273: The three parasitoid species (including Order and Family) names were added to the text

Reviewer 2 Report

The manuscript entitled “Predicting the invasion potential of the lily leaf beetle, Lilioceris lilii Scopoli (Coleoptera: Chrysomelidae), In North America” has been reviewed.

The authors have attempted to predict the capability of the lily leaf beetle, Lilioceris lilii Scopoli to establish on native North American Liliaceae plant species using a habitat suitability model. The latter was based on georeferenced occurrence records from the native, naturalized, and invasive range of this insect.

This is a scientifically sound research adopting correct methodology and data analyses. The results are interesting, original and well connected to Discussion, and a rich and relevant scientific literature was used throughout the text. I would suggest accepting the manuscript for publication after making the following few minor revisions:

L3 (title):  "Lilioceris lilii"  should be italicized

L4: replace  "In"  with  "in"

L4:  delete  " . "   after  "America"

L18 (Abstract):  change to  "if L. lilii"

L28 (Keywords): I'd suggest adding "Invasive insect" to the existing keywords to be as specific as possible

L31-32:  regarding info linked to the references "[1-3] and [4-6]" : which category of invasive species did you mean ? (animals or plants ??), please specify here

L36: once again as commented above: "to invasive species [7-10." ??

L39:  "new invaders [10-13]." ?? please specify 

L77:  replace "this new"  with  "this invasive" 

L83: delete " (. "

L90-91: 100 Km2

L97:  add  "in"   before  "North America"

L131:  replace  "where the beetle"   with   "where the presence of the beetle"

L150:  "South-East Regional" 

PAGE 5: Figure 2 and its legend should be of better quality

L174: add the continent (Asia) to which belongs "Mongolia"

L192 to L201: for a better presentation of your data, please indicate the country (USA or Canada) corresponding to each of the mentioned states or regions

L216:   attack by L. lilii

L219:  indicates this insect can

L232-233:  replace "Alternative chemical control mechanisms (e.g. neem) must"  with  "Biopesticides  (e.g. neem (azadirachtin)) as alternatives to chemical control must"

L234: please add the full names, Orders and families of the released species "the three parasitoid wasp species"

L243-244: replace the reference "(Orlova-Bienkowskaja 2013)"  by its corresponding number

L252:  northern China   (this is for consistency)

Author Response

Thank you for providing us feedback on our manuscript. I appreciate how quickly we received your review. After incorporating your suggestions, our article is much clearer and is rid of embarrassing mistakes.  

Below I address each of your suggested revisions. Before each response I added the new line number(s) that the corresponding revision can be found.

  • L3 (title):  "Lilioceris lilii"  should be italicized
    • L3: "Lilioceris lilii" was changed "Lilioceris lilii
  • L4: replace  "In"  with  "in"
    • L4: “In” was changed to “in”
  • L4:  delete  " . "   after  "America"
    • L4: The period after “America.” was removed
  • L18 (Abstract):  change to  "if L. lilii"
    • L31: “L. lilii” changed to “if lilii”
  • L28 (Keywords): I'd suggest adding "Invasive insect" to the existing keywords to be as specific as possible
    • L41: “Invasive insect” was added to the key words
  • L31-32:  regarding info linked to the references "[1-3] and [4-6]" : which category of invasive species did you mean ? (animals or plants ??), please specify here
    • L45: I added “The number of plant and animal species introduced” to clarify that I was talking about both invasive plant and animal species.
  • L36: once again as commented above: "to invasive species [7-10." ??
    • L52: I added “invasive plant and animal species” to clarify that I was talking about both invasive plant and animal species.
  • L39:  "new invaders [10-13]." ?? please specify 
    • L53 I added “plant and animal species,” to indicate that these models are being used by land managers to predict possible invasion by both plant and animal species.
  • L77:  replace "this new"  with  "this invasive" 
    • L94: I changed “this new” to “this invasive”
  • L83: delete " (. "
    • L99: “[22, 37-38](.” Changed to “[22, 37-38].”
  • L90-91: 100 Km2
    • L106-107: all “100km2” changed to “100km2
  • L97:  add  "in"   before  "North America"
    • L113: changed “North America” to “in North America”
  • L131:  replace  "where the beetle"   with   "where the presence of the beetle"
    • L148-149Changed “where the beetle” to “where the presence of the beetle”
  • L150:  "South-East Regional" 
    • L168: Changed “South East” to “South-East”
  • PAGE 5: Figure 2 and its legend should be of better quality
    • I added a title to Figure 2, and moved the variable names into the figure. The text is now larger and hopefully easier to read.
  • L174: add the continent (Asia) to which belongs "Mongolia"
    • 194: I added “(Asia)” after “Mongolia”
  • L192 to L201: for a better presentation of your data, please indicate the country (USA or Canada) corresponding to each of the mentioned states or regions
    • L214-225: I added USA or Canada after each mentioned state or territory
  • L216:   attack by  lilii
    • L251 I removed “the” from attack by the lilii
  • L219:  indicates this insect can
    • L254: I changed “the beetle can” to “this insect can”
  • L232-233:  replace "Alternative chemical control mechanisms (e.g. neem) must"  with  "Biopesticides  (e.g. neem (azadirachtin)) as alternatives to chemical control must"
    • L267-269: Changed to “Biopesticides (e.g. neem (azadirachtin)) used as an alternative to chemical control mechanisms”
  • L234: please add the full names, Orders and families of the released species "the three parasitoid wasp species"
    • L271-273: The names of the three parasitoid species (including Order and Family) were added to the text
  • L243-244: replace the reference "(Orlova-Bienkowskaja 2013)"  by its corresponding number
    • L283: Changed (Orlova-Bienkowskaja 2013) to [37]
  • L252:  northern China   (this is for consistency)
    • L292: “Northern China” changed to “northern China”

Reviewer 3 Report

Assessment of the manuscript entitled “Predicting the invasion potential of the lily leaf beetle, Lilioceris lilii Scopoli (Coleoptera: Chrysomelidae), In North America” for Insects (insects-903707)

Main comments

In this paper, the authors constructed a species distribution model for a coleopteran pest (Lilioceris lilii Scopoli (Coleoptera: Chrysomelidae)) in North America using maximum entropy with Maxent. This species has a broad distribution in Asia and Europe and some studies have suggested it can establish throughout much of North America. Thus, the authors aimed to test this prediction using a species distribution model to find suitable areas for this species. The study has a quite good approach since invasive species are among the global threats to ecosystems due to impacts on native flora and fauna worldwide.

After my assessment, my opinion is that this paper is well written, clear, and objective to answer a simple question, with the potential to help solve problems on a wide spatial scale about an invasive species. The analytical approach is well described and suitable for the proposed objective. I have only minor issues to address, as follows.

I congratulate the authors for a good piece of work.

Minor issues

Line 3. Species names should be italicized (Lilioceris lilii).

L 4. Change “In” to “in”. The endpoint could be removed.

L 83. Remove the parenthesis here “38] (.”

L 90-91. Correct: “100km2” (×3)

L 97. Change to “occurrences in North America”

L 100. Correct here “km2”

Figure 2. I think the variable names should be added to the figure title.

L 169. I think there is an unnecessary space here “model.   The”

Figure 3 has margins, Figure 4 does not. Shouldn’t they be similar in the presentation?

L 210-211. “spp.” should not be italicized.

Author Response

Thank you for providing us feedback on our manuscript. I appreciate how quickly we received your review. After incorporating your suggestions, our article is much clearer and is rid of embarrassing mistakes.  

Below I address each of your suggested revisions. Before each response I added the new line number(s) that the corresponding revision can be found.

  • Line 3. Species names should be italicized (Lilioceris lilii).
    • L3: "Lilioceris lilii" was changed "Lilioceris lilii
  • L 4. Change “In” to “in”. The endpoint could be removed.
    • L4: “In” was changed to “in”
  • L 83. Remove the parenthesis here “38] (.”
    • L99: “[22, 37-38](.” Changed to “[22, 37-38].”
  • L 90-91. Correct: “100km2” (×3)
    • L106-107: all “100km2” changed to “100km2
  • L 97. Change to “occurrences in North America”
    • L113: changed to “occurrences in North America”
  • L 100. Correct here “km2”
    • L116 : “km2” changed to “km2
  • Figure 2. I think the variable names should be added to the figure title.
    • I may not have understood what you were asking for. However, I added a title to Figure 2 and moved the variable names into the figure. The text is now larger and hopefully easier to read. Please let me know if this is a suitable change.
  • L 169. I think there is an unnecessary space here “model.   The”
    • L188: Unnecessary space removed from “model. The”
  • Figure 3 has margins, Figure 4 does not. Shouldn’t they be similar in the presentation?
    • I removed the boxes around picture “a” and “b” in Figure 3, so this figure will have the same format as Figure 4.
  • L 210-211. “spp.” should not be italicized.
    • L245-245: All “spp.” are no longer italicized